nanotechnology/green chemistry/materials science

zeolites, hierarchical materials, catalysis, selective oxidation

**Authors for correspondence:**
Xiaolei Fan
e-mail: xiaolei.fan@manchester.ac.uk
Christopher M. A. Parlett
e-mail: christopher.parlett@manchester.ac.uk

A contribution to 'Catalysis for a sustainable future' special collection Invited.
This article has been edited by the Royal Society of Chemistry, including the commissioning, peer review process and editorial aspects up to the point of acceptance.

# Palladium-doped hierarchical ZSM-5 for catalytic selective oxidation of allylic and benzylic alcohols

Shengzhe Ding[1], Muhammad Ganesh[1], Yilai Jiao[2], Xiaoxia Ou[1], Mark A. Isaacs[3,4], Mark S'ari[5], Antonio Torres Lopez[1,6], Xiaolei Fan[1] and Christopher M. A. Parlett[1,6,7,8]

[1]Department of Chemical Engineering and Analytical Science, School of Engineering, University of Manchester, Manchester M13 9PL, UK
[2]Shenyang National Laboratory for Materials Science, Institute of Metal Research, Chinese Academy of Sciences, 72 Wenhua Road, 110016 Liaoning, People's Republic of China
[3]Department of Chemistry, University College London, London WC1E 6BT, UK
[4]HarwellXPS, Research Complex at Harwell, Rutherford Appleton Laboratory, Harwell, Oxfordshire OX11 0FA, UK
[5]Nanoscience and Nanotechnology Facility, School of Chemical and Process Engineering, University of Leeds, Leeds LS2 9JT, UK
[6]Catalysis Hub, Research Complex at Harwell, Rutherford Appleton Laboratory, Harwell, Oxfordshire OX11 0FA, UK
[7]University of Manchester at Harwell, Diamond Light Source, Harwell Science and Innovation Campus, Didcot, Oxfordshire OX11 0DE, UK
[8]Diamond Light Source, Harwell Science and Innovation Campus, Didcot, Oxfordshire OX11 0DE, UK

SD, 0000-0003-2822-3882; MAI, 0000-0002-0335-4272; AT, 0000-0001-7378-1811; XF, 0000-0002-9039-6736; CMAP, 0000-0002-3651-7314

Hierarchical zeolites have the potential to provide a breakthrough in transport limitation, which hinders pristine microporous zeolites and thus may broaden their range of applications. We have explored the use of Pd-doped hierarchical ZSM-5 zeolites for aerobic selective oxidation (selox) of cinnamyl alcohol and benzyl alcohol to their corresponding aldehydes. Hierarchical ZSM-5 with differing acidity (H-form and Na-form) were employed and compared with two microporous ZSM-5 equivalents. Characterization of the four catalysts by X-ray diffraction, nitrogen porosimetry, $NH_3$ temperature-programmed desorption, CO chemisorption, high-resolution scanning transmission electron microscopy, X-ray photoelectron spectroscopy and X-ray absorption spectroscopy allowed investigation of their porosity, acidity, as well as Pd active sites.

The incorporation of complementary mesoporosity, within the hierarchical zeolites, enhances both active site dispersion and PdO active site generation. Likewise, alcohol conversion was also improved with the presence of secondary mesoporosity, while strong Brønsted acidity, present solely within the H-form systems, negatively impacted overall selectivity through undesirable self-etherification. Therefore, tuning support porosity and acidity alongside active site dispersion is paramount for optimal aldehyde production.

## 1. Introduction

The selective oxidation (selox) of alcohols plays a key role in synthesizing carbonyl species and finds widespread applications in chemical industries, including fragrancies, flavourings, pharmaceutical and agrochemical sectors [1]. In particular, the selox of benzylic and allylic alcohols to their carbonyl or acid derivatives is of high value to the food and fine chemical sectors [2]. Traditionally, alcohol selox has entailed the use of stoichiometric oxidizing agents, including permanganates [3], chromates [4] or $H_2O_2$ [5]. However, such heavy metal oxidants generate high volumes of aqueous metal salt waste, increasing economic and environmental impact. Meanwhile, peroxo-oxidants possess a potential risk from explosion, especially when deployed at large scale.

Therefore, the development of catalytic selox processes is of considerable interest, as these can satisfy several of the principles of green chemistry. Supported noble metals (such as Pd and Pt) have been employed as heterogeneous catalysts, with the ability to activate oxygen, either pure or from air, as oxidizing agent under mild operating conditions [6]. Therefore, such catalytic aerobic selox reactions represent a significant reduction in environmental impact through increased atom-economy and reduced e-factor while also reducing energy consumption relative to traditional stoichiometric oxidants [7].

There has been significant development of supported Pd-based systems for benzylic and allylic selox, with a degree of focus on the employment of mesoporous supports. Lu et al. [8] dispersed sub-nm Pd (less than 1 nm) on polyacrylontrile-functionalized mesoporous carbon to fix Pd cations. The 3 wt% Pd/C catalyst gave rise to 82.5% conversion of cinnamyl alcohol at 80°C in 6 h with 99% selectivity for cinnamyl aldehyde [8]. Shang et al. [9] reported a Pd single-atom catalyst by using $Al^{3+}$-enriched $Al_2O_3$ to anchor Pd, which resulted in 92% cinnamyl alcohol conversion at 80°C in 8 h, and exhibited a 15-fold increase in turnover frequency (TOF) compared to supported Pd nanoparticles (approx. 5 nm). X-ray absorption near edge structure (XANES) demonstrated the single-atom Pd sites to be positively charged (presenting as PdO), while nanoparticulate Pd was mainly metallic Pd [9]. We have also proposed a positively charged active species, namely PdO, for Pd catalysed allylic alcohol selox. We have demonstrated that increasing surface PdO, from increasing surface-to-bulk ratio with decreasing Pd size, exhibits significant enhancements in selox activity [10]. Operando synchronous DRIFTS/MS/X-ray absorption spectroscopy (XAS) further verifies PdO as the active species in allylic alcohol selox [11]. Incorporating complementary macropores, to generate hierarchical macroporous–mesoporous SBA-15, can further enhance performance [12]. For relatively small substrates (C4 and C9), this arises due to increasing active site density, while larger substrates also benefit from increased accessibility.

Zeolites are crystalline microporous aluminosilicates, in which a three-dimensional framework is formed by the corner-sharing $AlO_4$ and $SiO_4$ tetrahedral units [13]. They possess a range of desirable properties, such as ion-exchanging ability, high surface area and tuneable acidity. These result in widespread applications, including water treatment, gas separation and catalysis [14], and they have been employed as supports in selox of alcohols over supported Pd [15–17]. Li et al. [15] prepared a system via ion exchange of $Na^+$ with $Pd^{2+}$, generating catalysts with 2.8 nm Pd nanoparticles with optimal performance for benzyl alcohol selox with 66% conversion at 100°C in 4 h. However, when the average Pd size decreased to 2 nm, the conversion reduced, although the opposite was observed for geraniol or 2-octanol selox, where the small Pd nanoparticles outperformed the large equivalents [15].

One major drawback of zeolites is their microporous nature, which can hinder mass diffusion leading to coking and reduced in-pore diffusion of substrates. Incorporating complementary secondary porosity, either mesopores or macropores, has been demonstrated to help alleviate coking during gas-phase reactions [18–20]. It also can impart beneficial attributes to liquid-phase reactions through increased accessibility of active sites [21]. Common approaches in hierarchical zeolite preparation include 'bottom-up' methods, based on templating strategies, and 'top-down' methods, via desilication and dealumination [14,18,20,22–24]. Desilication methods typically consist of Si extraction in aqueous alkali solutions, leading to additional internal larger cavities, and presents a facile approach to

mesopore generation. Milina *et al.* [25] reported that the conversion of alkylation between benzyl alcohol and toluene was up to 90% over a hierarchical ZSM-5, while only 7% over commercial microporous ZSM-5. Moreover, hierarchical zeolites also provide additional pores for metal particle deposition, extending the application of zeolites beyond petrochemicals and small-molecule activation [22]. Martens *et al.* [26] used Pt-doped hierarchical ZSM-22 for hydro-isomerization of nonadecane and pristane, the latter requiring a lower temperature to obtain 20% conversion compared to Pt-doped microporous ZSM-22 (180°C versus 230°C).

Here, we report an investigation of Pd-doped ZSM-5 for aerobic selox of alcohols. The Pd-doped microporous–mesoporous hierarchical ZSM-5 in H-form (acidic) and Na-form (relatively neutral), prepared via desilication, are compared with corresponding Pd-doped microporous ZSM-5 to elucidate the role of ZSM-5 pore structure as well as intrinsic acidity in Pd impregnation and selox activity.

# 2. Methodology

## 2.1. Experimental reagents

Commercial ZSM-5 zeolites ($NH_4$-form, Si/Al = 25 and 40) were purchased from Zeolyst International. The following chemicals were used directly without further purification: sodium hydroxide (NaOH, ACROS Organics, ≥ 99%), ammonium nitrate ($NH_4NO_3$, ACROS Organics, ≥ 98%), sodium aluminate ($NaAlO_2$, Sigma Aldrich, ≥ 99%), toluene (Fisher Scientific, ≥ 99%), cinnamyl alcohol (Sigma Aldrich, ≥ 98%), mesitylene (ACROS Organics, ≥ 99%), ethanol (Fisher Scientific, ≥ 99.8%) and tetraaminepalladium(II) nitrate solution ($Pd(NH_3)_4(NO_3)_2$, Sigma Aldrich, 10 wt%).

## 2.2. Preparation of ZSM-5 supports

$NH_4$-form ZSM-5 zeolite (Si/Al = 25) was calcined at 550°C for 5 h (ramp rate 5°C min$^{-1}$) to convert it to the H-form. The zeolite was denoted as HMic-Z.

The H-form ZSM-5 (Si/Al = 25) was ion-exchanged with an aqueous 0.2 M $Na_2CO_3$ solution at 60°C for 0.5 h (1 g zeolite per 30 ml solution). The solid was isolated by filtration and washed to pH 8 with deionized water. The solid was dried at 100°C for 6 h and calcined at 550°C for 4 h (ramp rate 5°C min$^{-1}$). The zeolite was denoted as NaMic-Z.

The desilicated ZSM-5 was prepared by the methods reported by Perez-Ramirez and co-workers [27]. $NH_4$-form ZSM-5 zeolite (Si/Al = 40) was calcined at 550°C for 5 h (ramp rate 5°C min$^{-1}$) to convert it to the H-form. Six grams of H-ZSM-5 was vigorously stirred in aqueous NaOH solution (0.2 M, 30 ml g$^{-1}$ of zeolite) at 65°C for 0.5 h. The solid was recovered by centrifugation and was washed with deionized water to pH 8. Ion exchange was carried out in an aqueous $NH_4NO_3$ solution (0.1 M, 30 ml g$^{-1}$ of zeolite) at 80°C for 3 h. The product was isolated by centrifugation and washed to a neutral pH. The $NH_4$ ion exchange process was repeated twice. The material was dried at 100°C for 6 h and calcinated at 550°C for 4 h (ramp rate 5°C min$^{-1}$). The zeolite was denoted as HDe-Z. The same procedure was employed with the omission of the $NH_4$ ion exchange step to produce a Na-form denoted as NaDe-Z.

## 2.3. Pd impregnation

The microporous and hierarchical ZSM-5 supports were functionalized with Pd by wet impregnation. One gram of the support was stirred in an aqueous $Pd(NH_3)_4(NO_3)_2$ solution, with salt concentration adjusted to give a 1 wt% Pd loading. The slurries were stirred vigorously at 25°C for 18 h before being heated to 50°C to dry. The solids were calcined in air at 500°C for 2 h (ramp rate 1°C min$^{-1}$), prior to reduction at 400°C for 2 h (ramp rate 10°C min$^{-1}$) under flowing $H_2$ (10 ml min$^{-1}$). Pd contents were confirmed by inductively coupled plasma-optical emission spectroscopy, with Pd loading of 1.03 wt% Pd/NaDe-Z, 0.96 wt% Pd/NaMic-Z, 1.02 wt% Pd/HDe-Z and 1.04 wt% Pd/HMic-Z.

## 2.4. Characterization

Powder X-ray diffraction (XRD) was conducted employing a Philips X'Pert-PRO theta-theta PW3050/60 diffractometer (480 mm diameter) with a PW3064 sample spinner and X'Celerator (2.122° active length) one-dimensional detector in Bragg–Brentano geometry using a Copper Line Focus X-ray tube with Ni k$\beta$

absorber (0.02 mm; $K_\beta = 1.392250$ Å) $K_\alpha$ radiation ($K_{\alpha1} = 1.540598$ Å, $K_{\alpha2} = 1.544426$ Å, $K\alpha$ ratio 0.5, $K_{\alpha\text{ave}} = 1.541874$ Å). An incident beam Soller slit of 0.04 rad, 2° fixed anti-scatter slit, incident beam mask of 10 mm and programmable automated divergence slit giving a constant illuminated length of 10.0 mm and receiving Soller slit of 0.04 rad were used. Data collections from 5 to 75° coupled 2theta/theta at 0.0334° step, 1.7 s per step, were undertaken. The data were analysed using Jade 6.5.

Nitrogen adsorption/desorption isotherms of materials were measured by using a Micromeritics ASAP 2000 analyser. The materials were degassed under vacuum at 350°C for 12 h prior to nitrogen adsorption at −196°C. BET surface area was calculated over 0.01–0.15 relative pressure, and the microporous area was calculated over 0.4–0.6 relative pressure.

Ammonia temperature-programmed desorption (NH$_3$-TPD) analysis was conducted with a Micromeritics AutoChem II 2920 chemisorption analyser. Catalysts (approx. 200 mg) were degassed and dried at 550°C for 1 h and then cooled to 50°C under He. A gas mixture of NH$_3$ in He (10% : 90%, 30 cm$^3$ min$^{-1}$) was then introduced to saturate the catalyst followed by He purge (60 cm$^3$ min$^{-1}$) at 50°C for 2 h to remove the physically adsorbed NH$_3$. Finally, NH$_3$-TPD was performed by heating the catalyst from 50 to 600°C at 10°C min$^{-1}$ under He (30 cm$^3$ min$^{-1}$), and the desorbed NH$_3$ was monitored by gas chromatography (GC) using a thermal conductivity detector.

CO pulse chemisorption was performed with a Quantachrome chemBET 3000 system to measure Pd metal surface area. Samples were degassed at 150°C for 1 h under He (20 ml min$^{-1}$) before reduction at 100°C for 1 h under H$_2$. The mild reduction condition can avoid additional particle sintering. The CO chemisorption analysis was carried out at room temperature with a CO/Pd stoichiometry of 1 : 1 [10].

X-ray photoelectron spectroscopy (XPS) was conducted with a Kratos Axis SUPRA using monochromated Al k$\alpha$ (1486.69 eV) X-rays at 15 mA emission and 12 kV HT (180 W) and a spot size/analysis area of $700 \times 300$ µm. Spectral fitting was conducted using CasaXPS v. 2.3.19PR1.0. The binding energy of each sample was calibrated based on the Si 2p peak at 103.4 eV.

Fluorescence Pd K-edge (24.35 keV) XAS was conducted at Diamond Light Source on beamline B18 using a Si [3 1 1] monochromator, Pt-coated mirrors and a Vortex multichannel fluorescence detector. Spectra were processed using Demeter v. 0.9.26. Athena was used for normalization, background subtraction and linear combination fitting of XANES, and Artemis for extended X-ray absorption fine structure (EXAFS) fitting. Reference spectra of a Pd foil and PdO were also collected.

High-resolution transmission electron microscopy (TEM) was conducted with an FEI Tecnai F20 FEG TEM operating at 200 kV equipped with an Oxford Instruments X-Max SSD EDX detector (10 nm spot size). High-angle annular dark-field scanning transmission electron microscopy (HAADF-STEM) images were recorded on an FEI Titan3 Themis G2 operated at an accelerating voltage of 300 kV, equipped with a field-emission gun (X-FEG) operating at an extraction voltage of 4.5 kV, a monochromator and an FEI Super-X 4-detector EDX system.

## 2.5. Aerobic selective oxidation of alcohols

Catalyst screening was conducted in a two-neck 50 ml round-bottom glass flask at 90°C fitted with a water condenser. Cinnamyl alcohol (or benzyl alcohol) (1.1 g, 8.4 mmol) was dissolved in toluene (10 ml) with mesitylene (0.1 ml) employed as an internal standard. The reaction solution was heated to 90°C with agitation at 800 r.p.m. The catalyst (50 mg) was added to the reactor. Periodic reaction sampling was conducted by withdrawing an aliquot (0.25 ml). The aliquots were filtered into a vial and diluted with toluene (1.75 ml). The diluted extracted solutions were analysed by GC.

# 3. Results and discussion

## 3.1. Catalyst support properties

The properties of the zeolite supports were evaluated to confirm the retention of the zeolite structure in the hierarchal zeolites and compare physico-chemical properties of HMic-Z, NaMic-Z, HDe-Z and NaDe-Z. As presented in figure 1a, XRD patterns for all four samples show well-resolved peaks in the range of 5–75°, which are indexed to the characteristic pattern of the MFI zeolite structure (JCPDS data, card no. 01-084-0385), and confirm the retention of the zeolite cage structure within the hierarchical desilicated systems with long-range crystal ordering preserved [27].

As shown in figure 1b, nitrogen porosimetry reveals that HMic-Z and NaMic-Z both exhibit Type I isotherms, reflecting monolayer adsorption. According to IUPAC, this corresponds to microporous

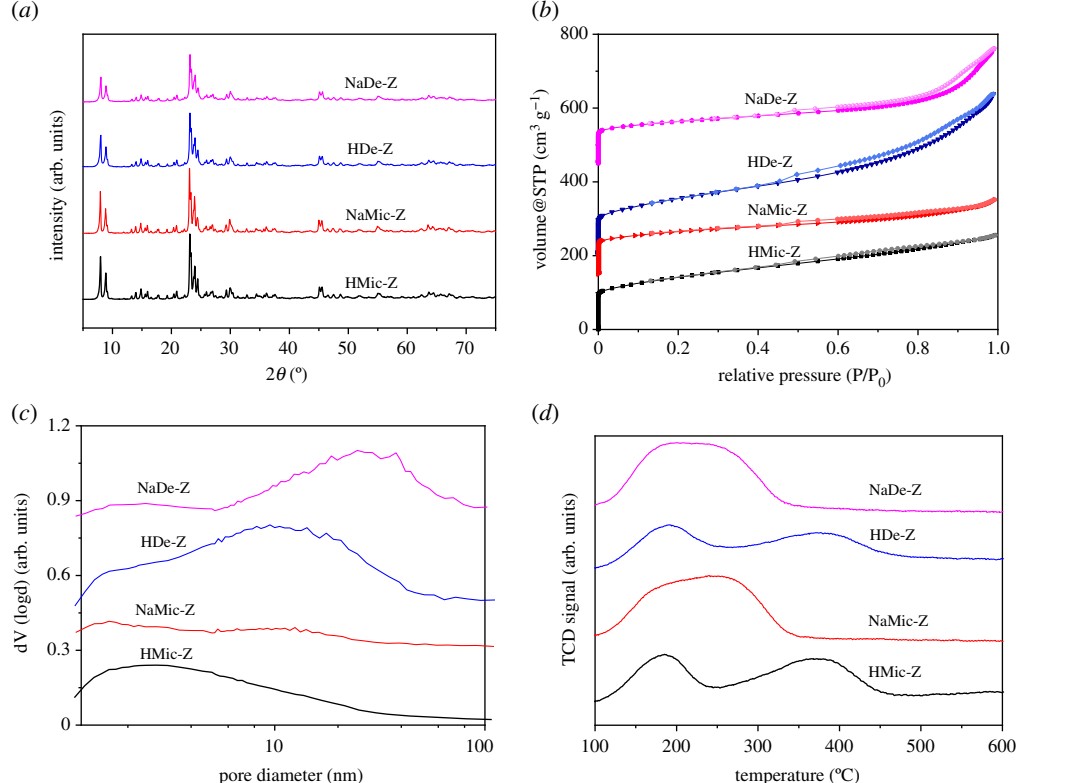

**Figure 1.** (*a*) Wide-angle XRD patterns, (*b*) nitrogen isotherms, (*c*) BJH pore size distributions and (*d*) ammonia-TPD of HDe-Z, HMic-Z, NaDe-Z and NaMic-Z.

**Table 1.** Properties of ZSM-5 (1 wt% Pd/ZSM-5) by nitrogen porosimetry.

| sample | BET surface area ($m^2 g^{-1}$)[a] | microporous area ($m^2 g^{-1}$)[b] | pore volume ($cm^3 g^{-1}$) | pore size (nm)[c] |
|---|---|---|---|---|
| HMic-Z | 491 ± 49 | 313 ± 31 | 0.39 ± 0.04 | <1 |
| HDe-Z | 537 ± 54 | 202 ± 20 | 0.68 ± 0.07 | ~15 |
| NaMic-Z | 410 ± 41 | 315 ± 32 | 0.31 ± 0.03 | <1 |
| NaDe-Z | 407 ± 41 | 228 ± 23 | 0.48 ± 0.05 | ~25 |

[a]Errors evaluated by repeated analysis of samples.

[b]Based on *t*-plot method.

[c]Based on BJH adsorption model with a Faas correction.

solids with relatively small external surfaces, such as zeolites [28]. By contrast, HDe-Z and NaDe-Z present Type IV isotherms with a characteristic hysteresis loop due to the capillary condensation occurring in mesopores. BJH pore size distributions (figure 1*c*) show mesopores in HDe-Z and NaDe-Z concentrated at around 15 and 25 nm, respectively, which are considerably larger and more abundant than pores within HMic-Z and NaMic-Z. Quantitative physical properties from nitrogen porosimetry are reported in table 1. Comparison of BET surface areas of hierarchical mesoporous–microporous desilicated ZSM-5 against microporous ZSM-5, either H form or Na form, shows both exhibit similar total surface area. However, there is a significant decrease in micropore surface areas for the hierarchical support materials, which indicates that a degree of the micropores is converted into mesopores during desilication. This is also apparent from the increased pore volume. We proposed that these mesopores will prove beneficial during the subsequent catalysis through enhanced accessibility of reactants to the active sites and shortened micropore diffusion path length [29]. Comparing the two hierarchical supports, HDe-Z possesses a higher mesopore area than NaDe-Z (335 versus 179 $m^2 g^{-1}$). This may arise from the dissolution of Al debris (resulting from desilication) during the $NH_4NO_3$ step [30] and may also explain the mesopore size distribution differences.

The acidity of the support materials was probed by NH$_3$-TPD, as illustrated in figure 1$d$. Both HMic-Z and HDe-Z show a peak at approximately 350°C, indicating strong acidity from the Brønsted acid sites of Si–O(H$^+$)–Al groups of zeolite framework [27]. HMic-Z shows a slightly greater intensity as the desilication process inevitably disrupts part of the zeolite framework. By contrast, the acid strengths of the two Na-form ZSM-5 are much weaker. The weak acidity in the four catalysts resulted from extra-framework Al, which yields predominately Lewis acid sites [30]. Abad *et al.* deposited Au onto nanocrystalline CeO$_2$ to generate Lewis acidity and evaluated the effect for aerobic selox of 3-octanol [31]. Here, Lewis acidity enhanced the TOF from 130 to 420 h$^{-1}$. However, to our knowledge, there is no evidence that the Lewis acidity in extra-framework Al can likewise improve Pd-catalysed selox, and given the comparable levels of Lewis acidity within our systems is a parameter we are unable to assess here.

## 3.2. Pd-doped zeolite characterization

The four zeolite supports were impregnated with 1 wt% palladium to produce a series of ZSM-5-based selox catalysts. Powder XRD patterns (electronic supplementary material, figure S1) show retention of the parent zeolite framework, and thus the impregnations and thermal processing showed no negative impact. An additional peak at 40°, corresponding to the Pd(111) reflection, is apparent in all samples except 1 wt% Pd/NaDe-Z. The absence indicates average Pd particle size of less than 2 nm [32]. Scherrer volume-average particle sizes of the Pd nanoparticles for the other three samples are reported in table 2. For comparison, Pd particle sizes estimated by STEM and CO chemisorption are also provided. These confirm a general decrease in Pd size within hierarchical mesoporous–microporous zeolites and Na-form zeolites. However, we do point out that inherent limitations hinder all techniques. These are lower size detection limits by XRD, potential for inaccessibility to probe molecules within porous architectures during  CO chemisorption due to pore blockage, and minute sample size by STEM. Since only CO chemisorption reflects truly accessible Pd, this can be considered the more appropriate method for comparison of the four catalysts.

N$_2$ porosimetry of the Pd-doped ZSM-5 catalysts (electronic supplementary material, figure S2) shows isotherm and hysteresis (where present) types, along with pore size distributions, are generally unchanged after Pd impregnation. However, as shown in table 2, Pd incorporation results in a decrease in micropore surface area. Some blockage of micropores by palladium nanoparticles may be the cause of this [10]. To further investigate the porosity of the four Pd-doped zeolites, the high-resolution TEM images are shown in figure 2. The structures of ZSM-5 particles in Pd/NaMic-Z and Pd/HMic-Z are comparable to each other but not to the equivalent forms subjected to desilication. For these (Pd/NaDe-Z and Pd/HDe-Z), the presence of complementary intracrystalline mesopores is clear, yielding hierarchical mesoporous–microporous ZSM-5. In figure 2$b$, for Pd/HDe-Z, clear lattice fringes of the zeolite are apparent, thus zeolite crystallinity is preserved after selective extraction of Si from ZSM-5 framework and Pd impregnation, which concurs with wide-angle XRD.

The acidity of the Pd/ZSM-5 materials, assessed by NH$_3$-TPD, is reported in the electronic supplementary material, figure S3, and shows Pd impregnation had only a minor influence on acidity. While acid strength across all four zeolites is unaffected, the concentration of weak acid sites (less than 250°C) does reduce slightly in all four catalysts. Weak acid sites are attributed to extra-framework Al [33,34], so these might provide an anchoring site for a small proportion of Pd nanoparticles. However, this decrease is negligible, with a significant degree of weak acidity still present.

Given the significant difference in atomic mass between Pd and the support elements, the z contrasting nature of HAADF-STEM is ideal for observing Pd sites within the ZSM-5 frameworks. Representative images are shown in figure 3 and electronic supplementary material, figure S4, while EDX images with corresponding STEM images are presented in the electronic supplementary material, figure S5. Pd nanoparticles on the H form of the zeolites, both microporous only and hierarchical, displayed a high proportion of Pd nanoparticle diameters spanning 1–3 nm; there is also a small degree of highly sintered species (approx. 100 nm). Pd/NaMic-Z exhibits a slightly broad average size, ranging from 2 to 10 nm, while the Pd on the hierarchal equivalent showed smaller average particle sizes, centred at approximately 2 nm and spanning 1–7 nm. Large (greater than 20 nm) Pd agglomerations in the Na form are not witnessed. The optimal dispersion of Pd/NaDe-Z was further confirmed by CO chemisorption, as presented in table 2, and is consistent with the XRD results. Pd dispersion of the remaining catalysts follows the following trend: Pd/HDe-Z > Pd/NaMic-Z > Pd/HMic-Z. This suggests the open nature of the hierarchical zeolite enables either greater dispersion, in contrast with XRD (which may be skewed due to size limitations of the technique and the presence of a small proportion

**7**

**Table 2.** Physico-chemical properties of Pd/ZSM-5.

| sample | BET surface area (m² g⁻¹)[a] | microporous area (m² g⁻¹)[a,b] | Pd dispersion from CO chemisorption (%)[a] | Pd size from XRD (nm) | Pd size from CO chemisorption (nm)[a] | Pd size from STEM (nm)[c] |
|---|---|---|---|---|---|---|
| Pd/HMic-Z | 450 ± 45 | 290 ± 29 | 13.0% ± 0.7% | 3.2 | 3.8 ± 0.2 | 1.7 ± 0.5 |
| Pd/HDe-Z | 549 ± 55 | 142 ± 14 | 29.3% ± 1.5% | 7.8 | 1.7 ± 0.1 | 2.2 ± 0.6 |
| Pd/NaMic-Z | 431 ± 43 | 215 ± 22 | 22.3% ± 1.2% | 2.6 | 2.9 ± 0.2 | 4.5 ± 2.0 |
| Pd/NaDe-Z | 369 ± 37 | 117 ± 12 | 62.1% ± 3.1% | — | 0.6 ± 0.1 | 2.4 ± 1.2 |

aError evaluated by repeat analysis of samples.
bBased on t-plot method.
cError from standard deviation.

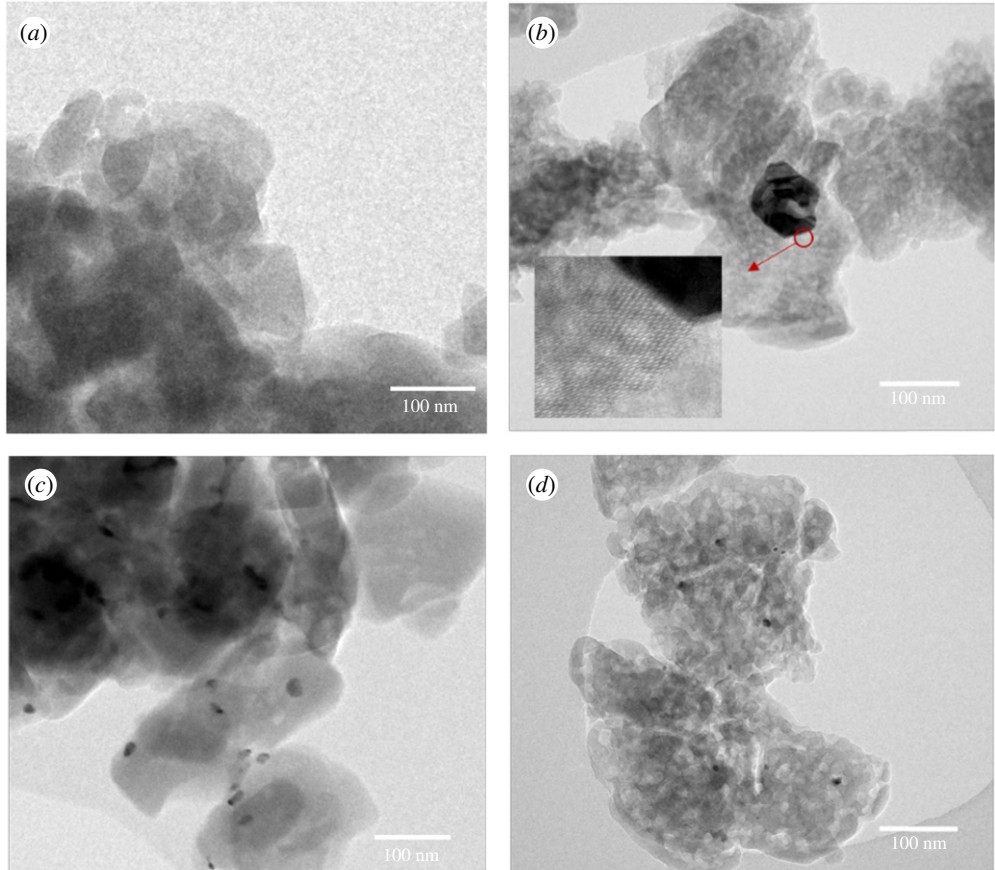

**Figure 2.** Representative TEM images of (*a*) Pd/HMic-Z, (*b*) Pd/HDe-Z, (*c*) Pd/NaMic-Z and (*d*) Pd/NaDe-Z.

of large Pd agglomerations), or greater accessibility. The low dispersion in the microporous-only systems could arise from a degree of inaccessible Pd from pore blockage.

Pd surface oxidation state was investigated by XPS (figure 4), with the Pd 3d doublet in all samples deconvoluted to two oxidation states, metallic ($Pd^0$ at 335.4 eV) and an oxidized state ($Pd^{2+}$ at 337.3 eV) [35]. $Pd^{2+}:Pd^0$, also presented in figure 4, shows Pd/HMic-Z to contain the highest $Pd^{2+}$ content, followed closely by Pd/HDe-Z and reflects the high degree of surface PdO formation on the small Pd nanoparticles, which increases with increasing surface-to-bulk ratio [10]. The Na forms show lower $Pd^{2+}$ content, which is inconsistent with the higher metal dispersion previously discussed. However, XPS will preferentially probe Pd on the external surface of the zeolite.

In contrast with XPS, XAS provides bulk characterization allowing evaluation of Pd on the external surface and within the pore framework. Normalized XAS spectra of the four catalysts are shown in figure 5, with XANES shown in the insert. The XANES linear combination fitting results, using reference PdO and Pd foil spectra, allowed the evaluation of $Pd^{2+}$ and $Pd^0$ content (reported in table 3). Here, we observed a correlation of PdO content with dispersion, with the hierarchical zeolite catalysts containing significantly higher PdO content, with the Na form displaying the highest proportion, while the two microporous-only zeolite systems possess comparable PdO density.

Further insight was gained from EXAFS, as shown in figure 6 and electronic supplementary material, table S1. The dominant scatterers in Pd/NaDe-Z are oxygen at 1.996 Å in the first shell and Pd in the second shell at 2.997 Å, consistent with the structure of PdO [36], although with reduced average coordination number (CN) of 2.8 and 1, respectively. A small contribution at 2.731 Å, attributed to metallic Pd, was also present, albeit with a CN much reduced from that of Pd foil (3.67 versus 12). These results were similar to those of our early work for highly dispersed Pd over Al-grafted SBA-15 [37], in which we considered the Pd nanoparticles to consist of a metallic core encapsulated by a PdO shell, surface oxidation arising due to exposure to air during storage. The low CNs here reflect minute average particle sizes in agreement with CO chemisorption. Pd/HDe-Z showed similar results, although with an increase in CN to 6.3 for the metallic phase, suggesting a larger metal core. Pd in the two microporous ZSM-5 samples was dominated by metallic Pd scatterers, with CN of the order

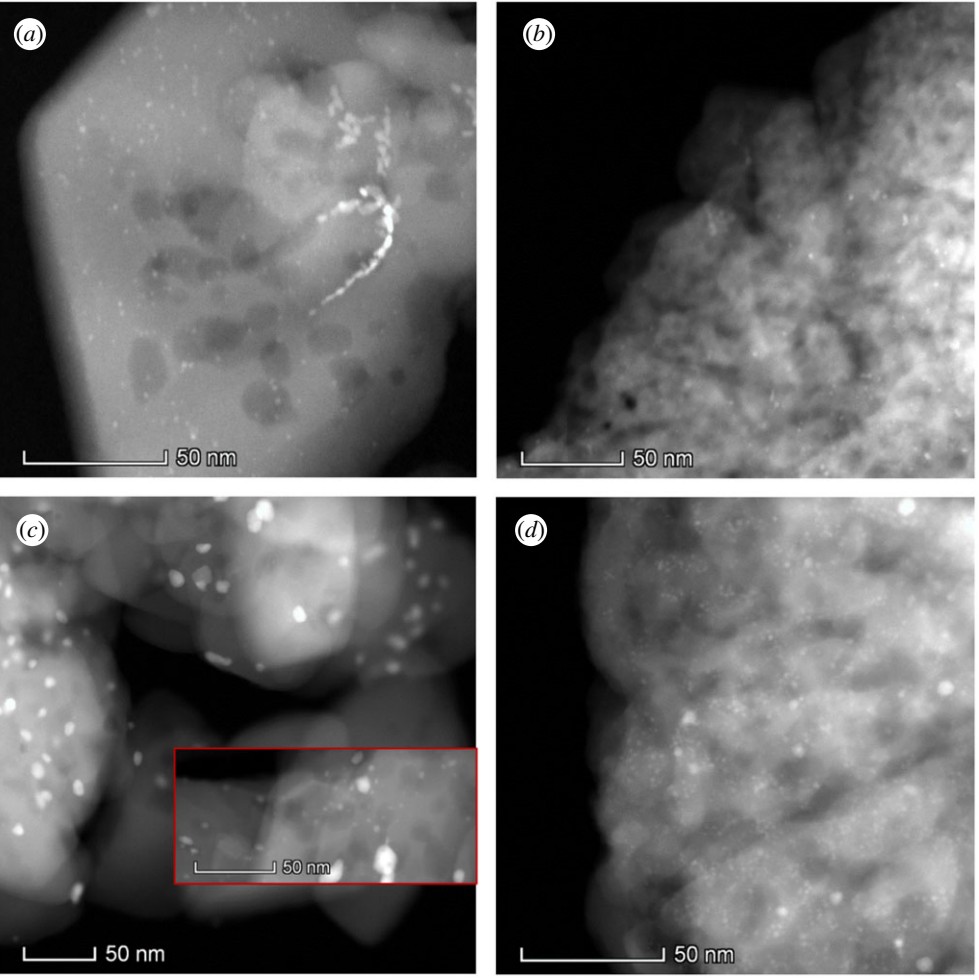

**Figure 3.** Representative high-magnification STEM images of (*a*) Pd/HMic-Z, (*b*) Pd/HDe-Z, (*c*) Pd/NaMic-Z and (*d*) Pd/NaDe-Z.

of 10 for both, 10.1 and 10.4 for H form and Na form, respectively, consistent with literature values for Pd-doped microporous zeolite [36,38,39]. Guillemot *et al.* [38] have proposed the relationship between first shell CN and nanoparticle size, from which we assess the diameter of Pd/HMic-Z and Pd/NaMic-Z to be 2.2 and 2.9 nm, respectively.

## 3.3. Alcohol selective oxidation

Cinnamyl alcohol selox was employed to screen and benchmark the four Pd/ZSM-5 catalytic materials. Figure 7 shows the introduction of complementary mesoporous structure within the hierarchical mesoporous–microporous zeolites, to impart enhanced conversion relative to the equivalent microporous catalyst; 1 wt% Pd/NaDe-Z showed the highest conversion of approximately 70% at 6 h and 60% selectivity to the desired aldehyde (electronic supplementary material, figure S6). The main by-product was hydrocinnamic acid (19%). This arises from over-oxidation of aldehyde and concurrent C=C hydrogenation. While hydrogenation under oxidizing conditions appears contradictory, under insufficient oxygen supply to the catalyst surface this has been previously observed over model Pd(111) [40] and 'real' Pd/SiO$_2$ systems [10].

While 1 wt% Pd/NaMic-Z resulted in similarly high aldehyde selectivity, conversion was servery limited. We attribute to the reduced accessibility of the Pd active sites within the micropore framework and reduced Pd dispersion according to CO chemisorption. The catalysts based on H-form zeolites displayed a similar enhancement with mesopore incorporation. However, cinnamaldehyde yields were much reduced, to around 20%, with the major product consisting of a high boiling point compound(s). GC-MS (electronic supplementary material, figure S7) suggested dicinnamyl ether formation, a previously reported by-product during cinnamyl alcohol Friedel–Crafts alkylation of toluene over zeolite Y [41]. To further investigate the etherification side reaction and the possibility of

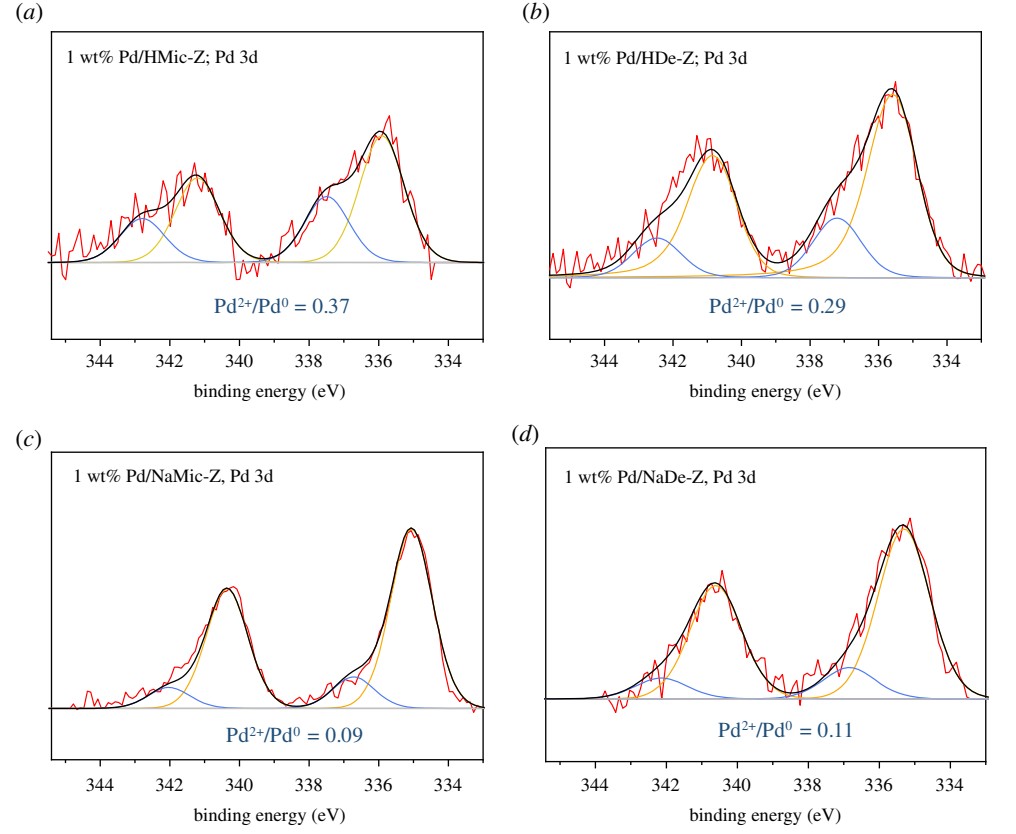

**Figure 4.** XPS profiles of Pd 3d regions, with deconvolution to Pd and PdO. (*a*) Pd/HMic-Z; (*b*) Pd/HDe-Z; (*c*) Pd/NaMic-Z; (*d*) Pd/NaDe-Z.

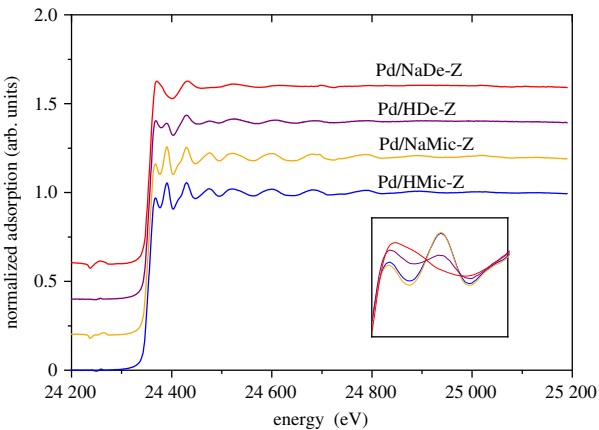

**Figure 5.** Normalized XAS spectra of Pd-doped ZSM-5.

**Table 3.** XANES linear combination fitting data for Pd-doped ZSM-5.

| sample | $Pd^{2+}$ | $Pd^0$ | $R_{factor}$ |
|---|---|---|---|
| Pd/NaDe-Z | $0.64 \pm 0.018$ | $0.36 \pm 0.018$ | $9.14 \times 10^{-4}$ |
| Pd/HDe-Z | $0.46 \pm 0.027$ | $0.54 \pm 0.027$ | $3.19 \times 10^{-3}$ |
| Pd/NaMic-Z | $0.18 \pm 0.020$ | $0.82 \pm 0.020$ | $3.86 \times 10^{-3}$ |
| Pd/HMic-Z | $0.23 \pm 0.015$ | $0.77 \pm 0.015$ | $3.84 \times 10^{-3}$ |

strong acid sites playing a role, 1 wt% Pd/HDe-Z was treated with $Na_2CO_3$ in mild condition to remove strong Brønsted acidity. As shown in the electronic supplementary material, figure S8, the capacity for cinnamaldehyde production was significantly enhanced and closely matches that of 1 wt % Pd/NaDe-Z. This results from switching off of etherification and thus indicating strong acidity to

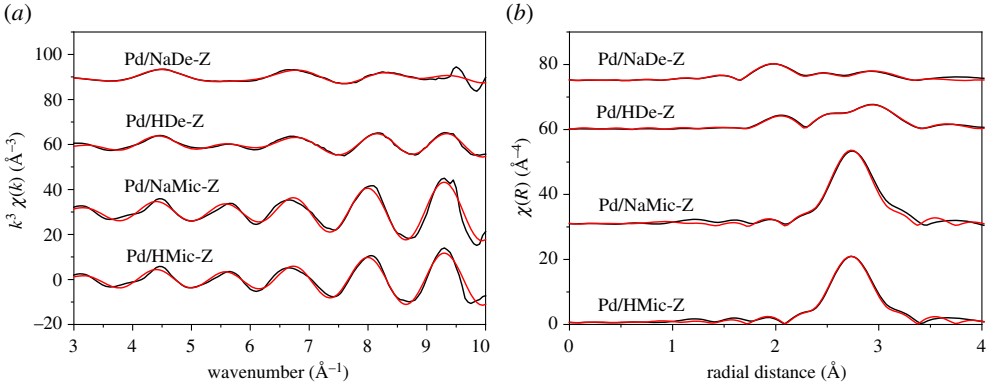

**Figure 6.** (a) $k^3$-Weighted Pd K-edge chi spectra and (b) radial distribution functions of Pd-doped ZSM-5 catalysts. The black curve represents experimental data and the red curve is the theoretical fit.

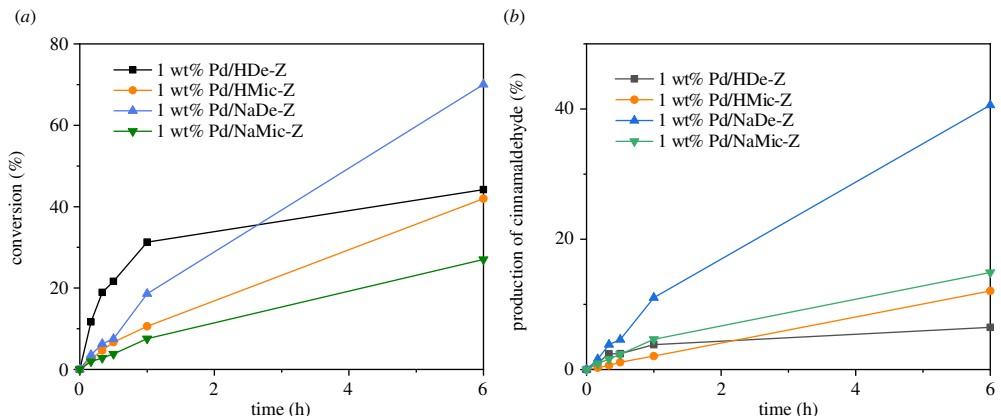

**Figure 7.** Selox of cinnamyl alcohol over different Pd-doped catalysts. (a) Conversion; (b) yield of cinnamaldehyde.

**Table 4.** Comparison of aldehyde productivity.

| | Pd/NaDe-Z | Pd/HDe-Z | Pd/NaMic-Z | Pd/HMic-Z |
|---|---|---|---|---|
| initial rate of cinnamaldehyde production (mmol $g_{Pd}^{-1}$ $h^{-1}$)[a] | $920 \pm 38$ | $412 \pm 15$ | $385 \pm 13$ | $177 \pm 6$ |
| initial rate of benzaldehyde production (mmol $g_{Pd}^{-1}$ $h^{-1}$)[a] | $904 \pm 38$ | $1153 \pm 43$ | $178 \pm 6$ | $507 \pm 18$ |

[a]Calculated over the initial 30 min.

be undesirable. XRD of the spent catalysts was employed to evaluate potential Pd agglomeration, with the resulting patterns shown in the electronic supplementary material, figure S9. These demonstrate no Pd sintering across all catalysts except Pd/NaMic-Z, which had a 20% decrease in full width at half maximum. This corresponds to an average Pd size of 3.3 nm.

Benzyl alcohol selox performance, as shown in the electronic supplementary material, figure S10, displays a similar enhancement with the incorporation of mesoporosity. Selectivity to the desired aldehyde is still favourable over systems without strong acidity, with benzaldehyde selectivity reaching approximately 90% for both 1 wt% Pd/NaDe-Z and 1 wt% Pd/NaMic-Z, compared to 50% for the two H-forms. The increased unsaturated aldehyde selectivity reflects a reduction in undesirable ether by-product formation. Etherification was again confirmed by GC-MS, identifying dibenzyl ether (electronic supplementary material, figure S11). This confirmation of the nature of the unknown high boiling point compounds, present in both reactions, indicates a competing reaction over the co-present strong Brønsted acids sites in HDe-Z and HMic-Z (electronic supplementary material, figure S3). Similarly, mesopores are also shown to escalate this undesired side reaction rate, which can again be attributed to superior internal substrate diffusion. Thus for optimal selox performance, strong zeolite acidity is undesirable, while complementary mesopores and highly dispersed Pd sites are desirable, the latter leading to enhanced nanoparticle surface PdO concentration, as shown in table 4.

# 4. Conclusion

The effects of ZSM-5 pore structure and acidity on deposited palladium nanoparticles and corresponding palladium-catalysed aerobic selox of alcohols have been investigated. After impregnation with Pd, the Na-form hierarchical ZSM-5 presented a high Pd dispersion, while the H-form also enabled high dispersion; they also resulted in a small degree of Pd agglomeration. Extensive characterization confirmed that mesopores in hierarchical ZSM-5 generally benefited Pd dispersion, which elevated PdO formation at the surface of small Pd nanoparticles. Previously, we have identified that surface PdO significantly boosts alcohol selox, with identical observations verified over ZSM-5 zeolites. Pd-doped hierarchical ZSM-5 exhibits optimal performance when benchmarked against microporous ZSM-5. Notably, H-form ZSM-5 resulted in significant self-etherification between alcohols, while Na-form ZSM-5 led to much higher selectivity to desirable unsaturated aldehyde.

Data accessibility. Additional data are available in the electronic supplementary material.

Authors' contributions. C.M.A.P. and X.F. conceived the project. C.M.A.P., X.F. and S.D. planned the experiments. S.D. and M.G. synthesized the supports and catalysts. S.D., M.G. and A.T.L. performed catalytic testing. S.D., M.A.I., Y.J., X.O., M.S. and C.M.A.P. conducted the materials characterization. S.D., C.M.A.P. and X.F. wrote the manuscript.

Competing interests. We declare we have no competing interests.

Funding. We thank the EPSRC National Facility for XPS (HarwellXPS), operated by Cardiff University and UCL, under contract no. PR16195, for XPS data collection. We thank Diamond Light Source and the UK Catalysis Hub for the provision of beamtime at B18 (UK Catalysis Hub SP15151).

Acknowledgements. This research has been performed with the use of facilities at the Research Complex at Harwell and the UK Catalysis Hub, including GC and chemisorption equipment. The authors would like to thank the Research Complex and the UK Catalysis Hub for access and support to these facilities and equipment. The authors thank Mark S'ari of Leeds EPSRC Nanoscience and Nanotechnology Facility (LENNF) for support and assistance in this work.

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
