## [Peer Review File · Royal Society Open Science]

Review History

RSOS-211086.R0 (Original submission)

Review form: Reviewer 1

Is the manuscript scientifically sound in its present form?

Yes

Are the interpretations and conclusions justified by the results?

Yes

Is the language acceptable?

Yes

Do you have any ethical concerns with this paper?

No

Have you any concerns about statistical analyses in this paper?

No

Recommendation?

Accept with minor revision (please list in comments)

Comments to the Author(s)

The paper reports the synthesis of Pd catalysts on both microporous zeolites and equivalents with additional mesoporous features for the selective oxidation of alcohols. The authors thoroughly characterize their materials and correlate these properties to catalytic activity demonstrating the beneficial effects of introducing these properties to the zeolite frameworks. As such I recommend this paper for publication after the following points have been addressed

- The authors report Pd particles sizes determined by three methods XRD, CO-chemisorption and STEM – these values agree with each to varying degrees depending on the sample. For instance, in comparing the HMiC-Z and HDe-Z the Pd particles sizes increase significantly if compared by XRD, decreases if measured by CO and remains constant (within errors I assume) by STEM. For the Na samples the Pd particle size decrease by CO and STEM. With these variations the authors should make a clear statement on which techniques they consider the most reliable and is most suitable to base their observations on. As in some cases different techniques are used to justify different activity trends.
- Standard deviations or errors could be estimate for each of the above methods (especially STEM) should be included.
- The authors should give additional information about the mechanisms behind the C=C hydrogenation under oxidising conditions. The text outlines an oxidative dehydrogenation however this would clean any surface hydrides to make H₂O?
- In the TPD data the authors do not mention the peaks at lower temperature corresponding to weak acidity and if they play a role in this reaction
- Pg 14 Line 39 (in submitted and compiled PDF) “aid” should be “air”
- The figure resolution should be addressed as in some cases they are a little pixelated.

Review form: Reviewer 2 (Lei Yu)

Is the manuscript scientifically sound in its present form?

Yes

Are the interpretations and conclusions justified by the results?

Yes

Is the language acceptable?

Yes

Do you have any ethical concerns with this paper?

Yes

Have you any concerns about statistical analyses in this paper?

No

Recommendation?

Major revision is needed (please make suggestions in comments)

Comments to the Author(s)

-Comments to the Authors

Thanks for inviting me to review the article named “Palladium-doped hierarchical ZSM-5 for catalytic selective oxidation of allylic and benzylic alcohols”. In this paper, the authors exploited explored the use of Pd-doped hierarchical ZSM-5 zeolites for aerobic selective oxidation (selox) of cinnamyl alcohol and benzyl alcohol to their corresponded aldehyde. The author found that the incorporation of complementary mesoporosity, within the hierarchical zeolites, enhances both active site dispersion and PdO active site generation. However, there are still some problems that need to be resolved. My advice is that this manuscript could be accepted by Royal Society Open Science if the following questions are properly answered.

1. In the Abstract, page1 line 48, there is an extra period.
2. Is there any difference in appearance between HDe-Z, HMic-Z, NaDe-Z and NaMic-Z? From XRD, there seems to be no difference.
3. It is recommended that the author do the EDX spectrum of Pd/HMic-Z, Pd/HDe-Z, Pd/NaMic-Z, Pd/NaDe-Z to clearly see the distribution and content of Pd on the zeolite.
4. In the reference section, authors need to use a uniform format.
5. How about the stability of the catalyst, will be the Pd nanoparticles agglomerated on the catalyst after the reaction. And how about the resistance of the catalyst to acids and bases.

Decision letter (RSOS-211086.R0)

Dear Dr Parlett:

Title: Palladium-doped hierarchical ZSM-5 for catalytic selective oxidation of allylic and benzylic alcohols

Manuscript ID: RSOS-211086

The editor assigned to your manuscript has now received comments from reviewers. We would like you to revise your paper in accordance with the referee and Subject Editor suggestions which can be found below (not including confidential reports to the Editor). Please note this decision does not guarantee eventual acceptance.

Please submit your revised paper before 11-Aug-2021. Please note that the revision deadline will expire at 00.00am on this date. If we do not hear from you within this time then it will be assumed that the paper has been withdrawn. In exceptional circumstances, extensions may be possible if agreed with the Editorial Office in advance. We do not allow multiple rounds of revision so we urge you to make every effort to fully address all of the comments at this stage. If deemed necessary by the Editors, your manuscript will be sent back to one or more of the original reviewers for assessment. If the original reviewers are not available we may invite new reviewers.

RSC Associate Editor:
Comments to the Author:
(There are no comments.)

RSC Subject Editor:
Comments to the Author:
(There are no comments.)

Reviewers' Comments to Author:

Reviewer: 1

Comments to the Author(s)

The paper reports the synthesis of Pd catalysts on both microporous zeolites and equivalents with additional mesoporous features for the selective oxidation of alcohols. The authors thoroughly characterize their materials and correlate these properties to catalytic activity demonstrating the beneficial effects of introducing these properties to the zeolite frameworks. As such I recommend this paper for publication after the following points have been addressed

- The authors report Pd particles sizes determined by three methods XRD, CO-chemisorption and STEM – these values agree with each to varying degrees depending on the sample. For instance, in comparing the HMiC-Z and HDe-Z the Pd particles sizes increase significantly if compared by

XRD, decreases if measured by CO and remains constant (within errors I assume) by STEM. For the Na samples the Pd particle size decrease by CO and STEM. With these variations the authors should make a clear statement on which techniques they consider the most reliable and is most suitable to base their observations on. As in some cases different techniques are used to justify different activity trends.

- Standard deviations or errors could be estimate for each of the above methods (especially STEM) should be included.

- The authors should give additional information about the mechanisms behind the C=C hydrogenation under oxidising conditions. The text outlines an oxidative dehydrogenation however this would clean any surface hydrides to make H₂O?

- In the TPD data the authors do not mention the peaks at lower temperature corresponding to weak acidity and if they play a role in this reaction

- Pg 14 Line 39 (in submitted and complied PDF) "aid" should be "air"

- The figure resolution should be addressed as in some cases they are a little pixelated.

Reviewer: 2

Comments to the Author(s)

-Comments to the Authors

Thanks for inviting me to review the article named "Palladium-doped hierarchical ZSM-5 for catalytic selective oxidation of allylic and benzylic alcohols". In this paper, the authors exploited explored the use of Pd-doped hierarchical ZSM-5 zeolites for aerobic selective oxidation (selox) of cinnamyl alcohol and benzyl alcohol to their corresponded aldehyde. The author found that the incorporation of complementary mesoporosity, within the hierarchical zeolites, enhances both active site dispersion and PdO active site generation. However, there are still some problems that need to be resolved. My advice is that this manuscript could be accepted by Royal Society Open Science if the following questions are properly answered.

1. In the Abstract, page1 line 48, there is an extra period.
2. Is there any difference in appearance between HDe-Z, HMic-Z, NaDe-Z and NaMic-Z? From XRD, there seems to be no difference.
3. It is recommended that the author do the EDX spectrum of Pd/HMic-Z, Pd/HDe-Z, Pd/NaMic-Z, Pd/NaDe-Z to clearly see the distribution and content of Pd on the zeolite.
4. In the reference section, authors need to use a uniform format.
5. How about the stability of the catalyst, will be the Pd nanoparticles agglomerated on the catalyst after the reaction. And how about the resistance of the catalyst to acids and bases.

Author's Response to Decision Letter for (RSOS-211086.R0)

See Appendix A.

RSOS-211086.R1 (Revision)

Review form: Reviewer 1

Is the manuscript scientifically sound in its present form?

Yes

Are the interpretations and conclusions justified by the results?

Yes

Is the language acceptable?

Yes

Do you have any ethical concerns with this paper?

No

Have you any concerns about statistical analyses in this paper?

No

Recommendation?

Accept as is

Comments to the Author(s)

The authors have addressed all minor comments.

Review form: Reviewer 2 (Lei Yu)

Is the manuscript scientifically sound in its present form?

Yes

Are the interpretations and conclusions justified by the results?

Yes

Is the language acceptable?

Yes

Do you have any ethical concerns with this paper?

No

Have you any concerns about statistical analyses in this paper?

No

Recommendation?

Accept as is

Comments to the Author(s)

The paper has been well revised and can be published in present form.

Decision letter (RSOS-211086.R1)

Dear Dr Parlett:

Title: Palladium-doped hierarchical ZSM-5 for catalytic selective oxidation of allylic and benzylic alcohols
Manuscript ID: RSOS-211086.R1

It is a pleasure to accept your manuscript in its current form for publication in Royal Society Open Science. The chemistry content of Royal Society Open Science is published in collaboration with the Royal Society of Chemistry.

RSC Associate Editor:
Comments to the Author:
(There are no comments.)

RSC Subject Editor:
Comments to the Author:
(There are no comments.)

Reviewer(s)' Comments to Author:

Reviewer: 2

Comments to the Author(s)

The paper has been well revised and can be published in present form.

Reviewer: 1

Comments to the Author(s)

The authors have addressed all minor comments.

Appendix A

Response to Referees

We thank the reviewers for their time, knowledgeable insight, and questions and comments. We have revised the manuscript and electronic supplementary information (ESI) to address these.

All amended sections of the manuscript and ESI are included and highlighted in yellow for clarity and identification.

Reviewer #1: The paper reports the synthesis of Pd catalysts on both microporous zeolites and equivalents with additional mesoporous features for the selective oxidation of alcohols. The authors thoroughly characterize their materials and correlate these properties to catalytic activity demonstrating the beneficial effects of introducing these properties to the zeolite frameworks. As such I recommend this paper for publication after the following points have been addressed:

1. The authors report Pd particles sizes determined by three methods XRD, CO-chemisorption and STEM – these values agree with each to varying degrees depending on the sample. For instance, in comparing the HMiC-Z and HDe-Z the Pd particles sizes increase significantly if compared by XRD, decreases if measured by CO and remains constant (within errors I assume) by STEM. For the Na samples the Pd particle size decrease by CO and STEM. With these variations the authors should make a clear statement on which techniques they consider the most reliable and is most suitable to base their observations on. As in some cases different techniques are used to justify different activity trends.

Response: We thank the reviewer for the comment. We provided sizes from a range of techniques due to the inherent limitation of each, that is, lower size detection limits by XRD, potential under measurements by CO chemisorption within porous architectures from pore blockage, and minute sample size by STEM. Given that CO chemisorption is the only technique to reflect size based on accessible Pd, hence it may be the most relevant measurement.

Action: Manuscript amended page 9

For comparison, Pd particles sizes estimated by STEM and CO chemisorption are also provided. These confirm a general decrease in Pd size within hierarchical mesoporous microporous zeolites and Na form zeolites. However, we do point out that inherent limitations hinder all techniques, that is, lower size detection limits by XRD, potential under measurements by CO chemisorption within porous architectures due to pore blockage, and minute sample size by STEM. Since only CO chemisorption reflects truly accessible Pd, this can be considered the more appropriate method for determining Pd sizes.

2. Standard deviations or errors could be estimate for each of the above methods (especially STEM) should be included.

Response: We thank the reviewer for the comment and have now included errors in all tables in the manuscript.

Action: Manuscript amended pages 8, 9, 13, and 16

Table 1. Properties of ZSM-5 (1 wt.% Pd/ZSM-5) by nitrogen porosimetry.

Sample	BET surface area (m ² g ⁻¹)	Microporous area (m ² g ⁻¹) ^a	Pore volume (cm ³ g ⁻¹)	Pore size (nm) ^b
HMic-Z	491 ± 49	313 ± 31	0.39 ± 0.04	<1
HDe-Z	537 ± 54	202 ± 20	0.68 ± 0.07	~ 15
NaMic-Z	410 ± 41	315 ± 32	0.31 ± 0.03	<1
NaDe-Z	407 ± 41	228 ± 23	0.48 ± 0.05	~ 25

^a based on *t*-plot method. ^b based on the BJH adsorption model with a Faas correction. Errors are evaluated by repeat analysis of a standard sample.

Table 2. Physicochemical properties of Pd/ZSM-5.

Sample	BET surface area (m ² g ⁻¹) ^b	Microporous area (m ² g ⁻¹) ^{a,b}	Pd dispersion CO-Chemisorption (%) ^b	Pd size from XRD (nm)	Pd size from CO-chemisorption (nm) ^b	Pd size from STEM (nm) ^c
Pd/HMic-Z	450 ± 45	290 ± 29	13.0% ± 0.7%	3.2	3.8 ± 0.2	1.7 ± 0.5
Pd/HDe-Z	549 ± 55	142 ± 14	29.3% ± 1.5%	7.8	1.7 ± 0.1	2.2 ± 0.6
Pd/NaMic-Z	431 ± 43	215 ± 22	22.3% ± 1.2%	2.6	2.9 ± 0.2	4.5 ± 2.0
Pd/NaDe-Z	369 ± 37	117 ± 12	62.1% ± 3.1%	-	0.6 ± 0.1	2.4 ± 1.2

^a based on *t*-plot method. ^b error evaluated by the repeat analysis of a standard. ^c error from standard deviation

Table 3. XANES Linear Combination Fitting data for Pd-doped ZSM-5

Sample	Pd ²⁺	Pd ⁰	R _{factor}
Pd/NaDe-Z	0.64 ± 0.02	0.36 ± 0.02	9.14 × 10 ⁻⁴
Pd/HDe-Z	0.46 ± 0.03	0.54 ± 0.03	3.19 × 10 ⁻³
Pd/NaMic-Z	0.18 ± 0.02	0.82 ± 0.02	3.86 × 10 ⁻³
Pd/HMic-Z	0.23 ± 0.02	0.77 ± 0.02	3.84 × 10 ⁻³

Table 4. Comparison of aldehyde productivity

	Pd/NaDe-Z	Pd/HDe-Z	Pd/NaMic-Z	Pd/HMic-Z
Initial rate of cinnamaldehyde production (mmol g _{Pd} ⁻¹ h ⁻¹) ^a	920 ± 38	412 ± 15	385 ± 13	177 ± 6
Initial rate of benzaldehyde production (mmol g _{Pd} ⁻¹ h ⁻¹) ^a	904 ± 38	1153 ± 43	178 ± 6	507 ± 18

3. The authors should give additional information about the mechanisms behind the C=C hydrogenation under oxidising conditions. The text outlines an oxidative dehydrogenation however this would clean any surface hydrides to make H₂O?

Response: We thank the reviewer for the comment and have revised the manuscript to clarify this potentially unpredicted pathway. We agree with the reviewer that under oxidising conditions, the liberated hydrogen from an oxidative dehydrogenation mechanism would yield H₂O, however, under oxygen-deficient conditions, there is insufficient O₂ supplied, and thus, surface hydrogen can remain. This is subsequently able to attack the C=C. We have demonstrated this phenomenon over model and real Pd catalysts in previous publications.

Action: Manuscript amended page 15

While hydrogenation under oxidising conditions appears contradictory, under insufficient oxygen supply to catalyst surface this has been previously observed over model Pd(111) [40] and “real” Pd/SiO₂ systems [10].

4. In the TPD data the authors do not mention the peaks at lower temperature corresponding to weak acidity and if they play a role in this reaction.

Response: We thank the reviewer for the comment and have addressed this in manuscript

Action: Manuscript amended page 8

The weak acidity in the four catalysts resulted from extra-framework Al, which yields predominately Lewis acid sites [30]. Abad et al. deposited Au onto nanocrystalline CeO₂ to generate Lewis acidity and evaluated the effect for aerobic selox of 3-octanol [31]. Here, Lewis acidity enhanced the TOF from 130 to 420 h⁻¹. However, to our knowledge, there is no evidence that the Lewis acidity in extra-framework Al can likewise improve Pd catalysed selox, and given the comparable levels of Lewis

acidity within our systems is a parameter we are unable to assess here.

5. Pg 14 Line 39 (in submitted and complied PDF) “aid” should be “air”.

Response: We thank the reviewer for pointing this typo out and have corrected the manuscript

Action: Manuscript amended page 14

in which we considered the Pd nanoparticles to consist of a metallic core encapsulated by a PdO shell, surface oxidation arising due to exposure to air during storage.

6. The figure resolution should be addressed as in some cases they are a little pixelated.

Response: We thank the reviewer for highlighting this issue and have replaced the figures with improved resolution.

Action: Manuscript amended throughout

Reviewer #2: Thanks for inviting me to review the article named “Palladium-doped hierarchical ZSM-5 for catalytic selective oxidation of allylic and benzylic alcohols”. In this paper, the authors exploited explored the use of Pd-doped hierarchical ZSM-5 zeolites for aerobic selective oxidation (selox) of cinnamyl alcohol and benzyl alcohol to their corresponded aldehyde. The author found that the incorporation of complementary mesoporosity, within the hierarchical zeolites, enhances both active site dispersion and PdO active site generation. However, there are still some problems that need to be resolved. My advice is that this manuscript could be accepted by Royal Society Open Science if the following questions are properly answered.

1. In the Abstract, page 1 line 48, there is an extra period.

Response: We thank the reviewer for pointing this typo out and have corrected the manuscript

Action: Manuscript amended page 1

allowed investigation of their porosity, acidity, as well as Pd active sites.

2. Is there any difference in appearance between HDe-Z, HMic-Z, NaDe-Z and NaMic-Z? From XRD, there seems to be no difference.

Response: Thank you for the query. While desilication does remove a proportion of the silica from the ZSM-5 framework, the majority of the zeolite crystallinity can remain intact, with this previously reported by Groen et al. [1], while Krisnandi et al. and Smail et al. have shown XRD patterns of H-form, Na-form, NH₄-form ZSM-5, to consist of very similar characteristic peaks [2, 3]. Therefore it is

not unexpected for the XRD pattern here to be very similar. In contrast, the local imaging by TEM/STEM clearly shows the presence of mesopores in HDe-Z and NaDe-Z, with N₂ porosimetry supporting this.

Action: Response to reviewer

3. It is recommended that the author do the EDX spectrum of Pd/HMic-Z, Pd/HDe-Z, Pd/NaMic-Z, Pd/NaDe-Z to clearly see the distribution and content of Pd on the zeolite.

Response: We thank the reviewer for the suggestion and have included EDX mapping of the four catalysts, while accurate Pd loading have been evaluated by ICP

Action: ESI amended page s5 and manuscript amended pages 5 and 11

Pd contents were confirmed by Inductively Coupled Plasma-Optical Emission Spectroscopy, with Pd loading of 1.03 wt.% Pd/NaDe-Z, 0.96 wt.% Pd/NaMic-Z, 1.02 wt.% Pd/HDe-Z, 1.04 wt.% Pd/HMic-Z.

Figure S1. Representative EDX images and corresponding STEM images of (a) (b) Pd/HMic-Z; (c) (d) Pd/HDe-Z; (e) (f) Pd/NaMic-Z; (g) (h) (l) Pd/NaDe-Z.

Representative images are shown in Figure 3 and Figure S4, while EDX images with corresponding STEM images are presented in Figure S5.

4. In the reference section, authors need to use a uniform format.

Response: We thank the reviewer for pointing out this error and have changed the style to Open Biology style according to the publication's requirement.

Action: Manuscript amended page 17 onwards

5. How about the stability of the catalyst, will be the Pd nanoparticles agglomerated on the catalyst after the reaction. And how about the resistance of the catalyst to acids and bases.

Response: We thank the reviewer for the comment regarding the stability of the Pd active sites and have evaluated this by XRD of the spent catalyst to investigate potential sintering. Of which none is apparent.

Action: Manuscript amended page 15 and ESI page S9

Figure S9. Overlaid XRD patterns of fresh and spent catalysts

XRD of the spent catalysts was employed to evaluate potential Pd agglomeration, with the resulting patterns shown in Figure S9. These demonstrate no Pd sintering across all catalysts except Pd/NaMic-Z, which had a 20% decrease in Full Width at Half Maximum. This corresponds to an average Pd size of 3.3 nm

Response: We thank the reviewer for their comment regarding resistance to acids and bases. Pore stability of zeolites in the presence of mineral acids and bases is widely known, with Al or Si leaching respectively. The latter being employed here to generate our hierarchical zeolites. So it can be expected that strong acids or bases would cause further destruction of the zeolite framework.

The presence of auxiliary alkali metals/bases is known in the literature to be critical in Au selox, but this is not the case for Pd and can lead to over oxidation during selox to give rise to the formation of the carboxylic acid. We have demonstrated the undesirable influence of Brønsted acidity on selox performance (albeit within our catalyst), and the addition of extra acidity would likely further enhance unwanted etherification reactions. Finally, the presence of weaker bases such as amine could act as capping agents, reducing accessibility to the Pd surface, however, they may also site-block the acid sites

within the zeolite and in the case of the protonated form catalyst yield similar improvement in aldehyde selectivity as witnessed in the Na forms.

While an interesting area for further investigation, we feel this is beyond the scope of this investigation and manuscript, and we feel it is inappropriate to speculate on.

Action: Response to reviewer

References

- 1 Groen, J., Peffer, L., Moulijn, J., Pérez-Ramirez, J. 2004 Mesoporosity development in ZSM-5 zeolite upon optimized desilication conditions in alkaline medium. *Colloids and Surfaces A: Physicochemical and Engineering Aspects*. **241**, 53-58.
- 2 Krisnandi, Y. K., Samodro, B. A., Sihombing, R., Howe, R. F. 2015 Direct synthesis of methanol BY partial oxidation of methane with oxygen over cobalt modified Mesoporous H-ZSM-5 catalyst. *Indonesian Journal of Chemistry*.
- 3 Smail, H. A., Rehan, M., Shareef, K. M., Ramli, Z., Nizami, A.-S., Gardy, J. 2019 Synthesis of uniform mesoporous zeolite ZSM-5 catalyst for friedel-crafts acylation. *ChemEngineering*. **3**, 35.